# AAV Mediated Delivery of Myxoma Virus M013 Gene Protects the Retina against Autoimmune Uveitis

**DOI:** 10.3390/jcm8122082

**Published:** 2019-11-29

**Authors:** Raela B. Ridley, Brianna M. Young, Jieun Lee, Erin Walsh, Chulbul M. Ahmed, Alfred S. Lewin, Cristhian J. Ildefonso

**Affiliations:** 1Department of Ophthalmology, University of Florida College of Medicine, Gainesville, FL 32610, USA; ridr0622@ufl.edu (R.B.R.); bmbowman6@gmail.com (B.M.Y.); erinmwalsh@ufl.edu (E.W.); 2Department of Molecular Genetics & Microbiology, University of Florida College of Medicine, Gainesville, FL 32610, USA; jl2683@mynsu.nova.edu (J.L.); ahmed1@ufl.edu (C.M.A.); lewin@UFL.EDU (A.S.L.)

**Keywords:** adeno-associated virus, uveitis, autoimmune, M013, myxoma

## Abstract

Uveoretinitis is an ocular autoimmune disease caused by the activation of autoreactive T- cells targeting retinal antigens. The myxoma M013 gene is known to block NF-κB (Nuclear Factor kappa-light-chain-enhancer of activated B cells) and inflammasome activation, and its gene delivery has been demonstrated to protect the retina against lipopolysaccharide (LPS)-induced uveitis. In this report we tested the efficacy of M013 in an experimental autoimmune uveoretinitis (EAU) mouse model. B10RIII mice were injected intravitreally with AAV (adeno associated virus) vectors delivering either secreted GFP (sGFP) or sGFP-TatM013. Mice were immunized with interphotorecptor retinoid binding protein residues 161–180 (IRBP_161–180_) peptide in complete Freund’s adjuvant a month later. Mice were evaluated by fundoscopy and spectral domain optical coherence tomography (SD-OCT) at 14 days post immunization. Eyes were evaluated by histology and retina gene expression changes were measured by reverse transcribed quantitative PCR (RT-qPCR). No significant difference in ERG or retina layer thickness was observed between sGFP and sGFP-TatM013 treated non-uveitic mice, indicating safety of the vector. In EAU mice, expression of sGFP-TatM013 strongly lowered the clinical score and number of infiltrative cells within the vitreous humor when compared to sGFP treated eyes. Retina structure was protected, and pro-inflammatory genes expression was significantly decreased. These results indicate that gene delivery of myxoma M013 could be of clinical benefit against autoimmune diseases.

## 1. Introduction

Myxoma virus is a member of the poxvirus family. This virus is known to cause myxomatosis in the European rabbit with a high mortality rate [1]. Although not a human pathogen, it is well known that myxoma virus replicates in human cancer cells and holds great clinical value as an oncolytic viral therapy [2,3,4]. The success of the myxoma virus infection relies in its multiple immunomodulatory genes which target signaling pathways needed by the host to sense its presence and initiate an immune response [5]. The M013 gene was characterized by Rahman et al. in 2009 as an inhibitor of the apoptosis-associated speck-like protein containing a CARD (ASC-1) and nuclear factor kappa-light-chain-enhancer of activated B cells (NF-κB) signaling molecules [6]. The deletion of this gene from the myxoma virus genome results in the activation of both the inflammasome and NF-κB signaling in infected cells leading to the stimulation of the immune system [7]. This dual function of M013 is carried out by different protein regions of M013. Garg et al. demonstrated that the C-terminus domain of M013 is required for its inhibition T-minus of the molecule affects its ability to bind and inhibit the function of ASC-1 [8].

The therapeutic value of M013 has been explored by our group using an endotoxin induced uveitis (EIU) mouse model. We have developed an AAV vector which delivers a secretable and cell-penetrating modified M013 [9]. In this vector, M013 gene is fused to an human immunodeficiency virus type 1 (HIV-1) Tat-peptide sequence which allows secreted molecules to penetrate cells from the extracellular space. We render the TatM013 secretable by fusing it to a GFP sequence which contained an Igκ secretion signal (sGFP), whose purpose is to both serve as a reporter gene and to induce the secretion of the TatM013 protein. We demonstrated that conditioned media containing secreted TatM013 can inhibit the induced secretion of interleukine-1 beta (IL-1β), a downstream target of ASC-1 and a gene whose expression is regulated by NF-κB. Furthermore, when this secreted TatM013 is delivered into the retina of the endotoxin-induced uveitis mouse model, it is capable of inhibiting LPS-induced ocular inflammation and reducing the retinal concentration of interleukine-1 beta (IL-1β). This secretable and cell-penetrating M013 allows inhibit of both ASC-1 and NF-κB signaling through the entire retina after an intravitreal injection, which has the benefit of potentially requiring lower titers of the vector to observe a protective effect. 

Inflammation is a natural process through which an organism protects itself against an invading pathogen or limits the effects of damaged cells. The eye is a recognized immune privileged organ that employs multiple mechanisms to maintain an immunosuppressive environment [10]. The anterior chamber associated immune deviation (ACAID) mediates the protection of the eye from the deleterious effects of an adaptive immune response [11]. However, the eye is still susceptible to inflammation caused by systemic diseases such as cancer [12], multiple sclerosis [13], juvenile idiopathic arthritis [14], and vascular diseases such as Bechet’s disease [15]. Autoimmune uveitis or uveoretinitis is the result of a systemic tolerance breakdown that leads to the generation of autoreactive T-cells (Th1 and Th17) which recognize retina specific antigens such as the interphotoreceptor retinoid binding protein (IRBP) or S-arrestin [16,17]. Studies in human blood cells have demonstrated that cytokines associated with the adaptive immune system such as IL-17A and IFN-γ play key roles in the pathogenesis of autoimmune uveitis [18]. However, complementary studies have also identified a role for elements of the innate immune response such as retinal microglia cells [19], dendritic cells [20], and the cytokine tumor necrosis factor α (TNF-α) [21]. Furthermore, animal studies have demonstrated that IL-1β signaling is essential for the establishment of that autoimmune uveitis, since its abrogation provides protection of the retina [22] while its supplementation during early stages leads to a more pronounced retinal damage [23]. These observations point towards the complex pathophysiology of this disease. 

Current therapies include the use of steroids, immunosuppressants, and biologicals such as adalimumab [24]. Although these therapies can curtail the retinal damage associated with the disease, their prolonged use pose other risks such as increased intraocular pressure or infection. Thus, there is a need to identify alternative therapies that can alleviate the disease with less risk. Hazards associated with current therapies can be overcome by developing a therapeutic agent that can be delivered locally and that can provide prolonged protection after a single ocular injection. Given the clinical success of the adeno-associated virus (AAV) vectors in the treatment of Leber’s Congenital Amaurosis [25], we developed an AAV vector that can deliver an anti-inflammatory gene through an intravitreal injection using an AAV capsid mutant known to transduce different retinal cells [26]. This approach is expected to provide local long-term anti-inflammatory effects after a single injection based on the long-term effects described for AAV-RPE65 in Leber Congenital Amaurosis type 2 (LCA2) patients [27].

There are multiple mouse models of autoimmune uveitis that recapitulate key features of the human disease. The most common model explores the susceptibility of the B10.RIII mouse strain towards a peptide from the interphotoreceptor retinoid binding protein (IRBP). Established by Rachel Caspi in 1988 [28], uveitis is induced by immunizing mice with an IRBP peptide emulsified in complete Freund’s adjuvant. Immunized mice develop a severe autoimmune response against the retina within two weeks after immunization which resolves over time with a slow progressive damage [17,29]. Other models involve the transfer of autoreactive T-cells or animals carrying a mutant T-cell receptor that causes the spontaneous development of autoimmune uveitis [16]. Finally, the characterization of an IRBP peptide capable of inducing moderate to severe uveitis in C57BL6J mouse strain opens the door to understand the molecular underpinnings of autoimmune uveitis by combining it with well characterized transgenic mice [30].

Herein, we have evaluated the efficacy of M013 retinal gene delivery via an AAV vector using a mouse model of experimental autoimmune uveoretinitis (EAU). This model combines the B10.RIII mouse strain which carries H2^r^ allele of the MHC (major histocompatibility complex) gene with the interphotoreceptor retinoid binding protein (IRBP) peptide 161–180 and a strong adjuvant to break systemic tolerance towards retinal antigens and induce autoimmune uveoretinitis [28]. The peak of inflammation in this mouse model takes place 14 days post-immunization and is characterized autoreactive Th1 and Th17 cells [29]. Therefore, we chose the 14 days time point to evaluate the effects of retina M013 expression. 

## 2. Experimental Section

### 2.1. Animals

C57BL6J and B10.RIII mice were obtained from Jackson Laboratory (Bar Harbor, ME, USA) and maintained in a pathogen free facility with a 12 hours light/12 hours darkness cycle on a normal diet with water ad libidum. Eight-week-old female mice were included in our studies due to their greater inflammatory response when compared to male littermates. All experiments were approved by the University of Florida Institutional Animal Care and Use Committee and adhere to the guidelines delineated by the Association for Research in Vision and Ophthalmology (ARVO) (UF IACUC 201809086).

### 2.2. Adeno-Associated Viral (AAV) Vector Production

We have previously described the generation of our AAV plasmid pTR-smCBA-sGFP-TatM013 [9]. This plasmid contains the terminal repeats (TRs) of AAV2 which flank the small chicken-beta acting (smCBA) promoter and the Igk leading sequence fused to the GFP cDNA (sGFP). We fused the sGFP with a HIV-Tat peptide sequence that is associated with the M013 cDNA tagged with a V5 epitope tag (TatM013v5). Finally, a furin cleavage site (FCS) was included between the sGFP and the TatM013v5 to allow the posttranslational separation of these proteins. 

We packaged this vector using a capsid mutant version of AAV2 named AAV2quad (Y-F)+T491V. This vector was characterized by Kay et al. who demonstrated its ability to transduce multiple retina cells after an intravitreal injection [26]. This vector was packaged by the Ocular Gene Therapy Core in the department of Ophthalmology following published methods [31]. Briefly, HEK293 cells were co-transfected with the pTR-smCBA-sGFP-TatM013 plasmid and the appropriate helper plasmids including the capsid mutant helper plasmids using linear polyethyleneimine (PEI~2500). Cells were grown in a cell factory until significant cytopathic effects were noticed. Cells were harvested and lysed to release viral particles. Viral particles were separated from the cellular debris by ultracentrifugation in an iodexanol gradient. Isolated particles were further purified and concentrated by FPLC. The vector preparation was resuspended in Hank’s balance salt solution as 25 µL aliquots. Titration of the vector was determined using quantitative PCR and the quality of the viral particles was determined by a sodium dodecyl sulfate polyacrylamide gel electrophoresis (SDS-PAGE) silver stain to detect the viral capsid proteins. Vector aliquots were stored at −80 °C until needed. 

### 2.3. Intravitreal Injection

Mouse eyes were dilated with topical drops of Tropi-Phen^®^ (2.5% Phenylephrine HCl, 1% Tropicamide ophthalmic solution; Baush and Lomb, Laval, QC, Canada). Mice were anesthetized with a mixture of ketamine/xylazine (100 mg/kg ketamine, 4 mg/kg xylazine). Anesthetized mice were placed on the platform of a Leica M165 stereoscope. An incision was made through the limbus using a 25 G needle. A Hamilton syringe (30 G 1.0 mL) was inserted carefully avoiding the lens capsule until it reached the vitreous body. A total of 1 µL of AAV vector diluted in normal saline and AK-Fluor^®^ (Fluorescein) was slowly injected. Afterwards the needle was removed, and the mouse eye was treated with Vetropolycin HC ophthalmic ointment^®^ (bacitracin-neomycin-polymyxin with hydrocortisone acetate 1%, Greeley, CO, USA). Mice received an intraperitoneal injection of atipamezole (0.625 mg/kg) to reverse the anesthesia and maintained on a heated plate until ambulatory. 

### 2.4. Electroretinogram

Mice were dark adapted for 12 h before conducting the electroretinogram (ERG). Mice were anesthetized with a ketamine/xylazine mixture (100 mg/kg ketamine, 4 mg/kg xylazine) and their eyes were dilated with two drops of an ophthalmic solution of Tropi-Phen^®^ (2.5% Phenylephrine HCl, 1% Tropicamide ophthalmic solution; Melborne, FL, USA). Once anesthetized, mice were placed on a heated platform and gold electrodes were placed on their corneas, and reference electrodes were placed in their mouth and tail base. To ensure proper contact between the electrodes and the cornea the eyes were covered with a drop of Genteal^®^ (hypermellose 0.25%, CVS Pharmacy, Gainesville, FL, USA). The retinas were stimulated by three flashes of 20 cds/cm^2^ using the Xenon lamp of a full field dome, with a 120 s separation between flashes. Recordings after each flash were averaged and the amplitudes (in mV) was recorded. The anesthetics effects in the mice were reversed by an intraperitoneal injection of atipamezole. Mice were maintained on a heated plate until they became ambulatory. 

### 2.5. Spectral Domain Optical Coherence Tomography

Mice were anesthetized as described for ERG with a ketamine/xylazine mixture. Their eyes were dilated with two drops of Tropi-Phen^®^ (Tonawanda, NY, USA). Anesthetized mice were then placed on a platform facing the SD-OCT camera. A drop of Genteal^®^ was added to the eyes to keep them from dehydrating. SD-OCT images were acquired using a Bioptigen Envisu R equipment (Leica Microsystems, Buffalo Grove, IL, USA). Once the image was centered on the optic nerve head, a total of 250 B-scan images were obtained for each eye and averaged into 25 high resolution B-scan images. 

### 2.6. Quantification of Infiltrative Cells within the Vitreous Humor Using SD-OCT Images

Quantification of infiltrative cells using SD-OCT images has been validated as an alternative to histological scoring of retinal inflammation [32]. B-scan images were opened with an ImageJ OCT plug. Three B-scan images per eye were duplicated. Each image was then inverted and flipped so that the lens capsule would be on the top of the image. The background was subtracted, and the image scale was set (1024 pixels = 1400 microns). We then auto adjusted the brightness/contrast and adjust the threshold to 11%. To separate cell clumps, we applied the watershed function. Finally, the vitreous body was selected using a polygon function to exclude the retina and the lens capsule. To count the number of particles in the selected area we used the count particles function with the following parameters: 50–300 microns^2^ and exclude on edges. The area between the lens capsule and the retina nerve fiber layer was selected and the outside area was cleared. The number of particles counted was then recorded. 

### 2.7. Fundoscopy

We injected mice with a mixture of ketamine/xylzaine as described above. Their retinas were dilated with two drops of Tropi-Phen^®^. Their whiskers were trimmed, and their eyes received a drop of Genteal^®^ to avoid the cornea from dehydration. Mice were placed on a platform and their eyes were aligned to a Micron III fundus camera. Images were taken once the optic nerve head was visible and were focused on the retina. Images were saved using the tiff format. Anesthetics were reversed with a dose of atipamezole as described above and mice were kept on a warm pad until ambulatory.

### 2.8. Clinical Scoring of Fundus Images

Fundus images were labeled using the animal cage number and ear punch. Images were scored by individuals unaware of the treatment. A clinical scoring system similar to that described by Xu et al. was used to quantify the magnitude of retina damage from digital fundus images [33]. Both individuals were provided with the following grading scale: 0 = No disease, 1 = Few infiltrating cells, 2 = Many infiltrating cells, 3 = 2 + engorged blood vessels, 4 = 3 + hemorrhage, 5 = 4 + retinal edema or detachment. Individuals registered their scores and reported them directly to the main investigator who used the analyzed data based on the average score and the treatment status.

### 2.9. Histological Evaluation

Mice were euthanized by CO_2_ asphyxiation. Eyes were gently enucleated and rinsed once with sterile phosphate buffered saline (PBS). Eyes were then transferred into a 1.5 mL containing 4% paraformaldehyde in PBS and incubated on ice for 30 min. Eyes were then transferred to a 2.0 mL tube containing 70% ethanol and stored at 4 °C. The eyes were then embedded in paraffin, sectioned, and stained with hematoxylin and eosin by Histology Tech Services. Briefly, 500 micrometers were cut off from the paraffin block through the cornea-optic nerve axis, followed by the capture of 8 step sections at 80 micrometer intervals. Finally, the slides were stained with hematoxylin and eosin following standard procedures. 

### 2.10. Retinal RNA Extraction and cDNA Synthesis

Mice were euthanized by CO_2_ asphyxiation followed by cervical dislocation. A slit through the cornea was made using a sharp surgical scalpel. Aqueous humor was blotted to an absorbent towel, and the eye was squeezed until the lens protruded out of the eye. After removing the lens, the retina was removed by further squeezing the ocular globe. The retina was placed in 400 µL of Trizol Reagent and homogenized using a motorized pestle by giving 20 firm strokes. Samples were then incubated at room temperature for 5 min followed by the addition of 100 µL of chloroform. Samples were mixed by vortexing for 15 s then incubated at room temperature for 2 min followed by a centrifugation at 18,000 × *g* for 15 min at 4 °C. The aqueous phase was carefully harvested, and an equal volume of 100% ethanol was added to the sample. Samples were then vortexed for 15 s and incubated at room temperature for 10 min. Another centrifugation at 18,000 × *g* for 15 min at 4 °C was performed. The aqueous phase was removed, and the RNA pellet was washed with 500 µL of 100% ethanol and mixing several times followed by a centrifugation as done in earlier steps. A second wash with 500 µL of 70% ethanol was performed and a final centrifugation at 18,000 × *g* for 20 min at 4 °C was done to precipitate the RNA pellet. The 70% ethanol was removed with a micropipette and the RNA pellet was air-dried for 15 min. RNA was solubilized in 100 µL of diethylpyrocarbonate (DEPC) treated water and heated to 65 °C for 15 min. 

RNA concentration was determined with a Qubit 3.0 fluorimeter (ThermoFisher, Waltham, MA, USA) using the Qubit RNA HS Assay kit (ThermoFisher, Waltham, MA, USA) as per the manufacturer’s protocol. A cDNA library was synthetized with 200 ng of total RNA using the iScript cDNA synthesis kit from Bio-Rad according to the manufacturer’s protocol. The cDNA concentration was then determined using the Qubit DNA HS assay kit and samples were then diluted to 1 ng cDNA/µL in nuclease free water. The cDNA was stored at −20 °C until needed.

### 2.11. Reverse Transcribed Quantitative PCR (RT-qPCR)

RT-qPCR reactions were performed using a total of 4 ng of cDNA from each sample. Reactions were made using the SsoAdvanced Universal SYBR Green Supermix (Bio-Rad, Hecules, CA, USA) according to the manufacturer’s protocol. Oligos sets for each gene of interest are described in Table 1. Quantitative RT-PCR (real time polymerase chain reaction) was performed using a CFX96 thermocycler from Bio-Rad using the conditions described on Table 2.

### 2.12. Retinal Protein Extraction and Protein Determination 

Mouse retinas were squeezed out of the eye globe and collected in 100 µL of NP-40 lysis buffer (50 mM Tris-HCl (pH 8.0), 150 mM NaCl, 1% NP-40) supplemented with 100 mM EDTA and Hank’s Protease Inhibitors cocktail. Retinas were homogenized using a motorized pestle with 20 firm strokes followed by sonication for 20 s. Homogenates were then centrifuged at 4 °C for 10 min at 12,000 × *g*. Clear lysate was collected and placed on a new 1.5 mL microcentrifuge tube.

### 2.13. Protein Determination

Protein concentration was determined using the Bio-Rad Dc Assay as per the manufacturer’s protocol. Briefly, 5 µL of either lysate, BSA standard (0.3–1.5 µg/µL), or NP-40 lysis buffer were placed on a 96-well plate. Each well then received 25 µL of reagent A’ (1 mL reagent A supplemented with 20 µL of reagent S). All wells then received 200 µL of reagent B and the plate was incubated at room temperature for 15 min on a plate shaker. Absorbance was read at 750 nm using a Clario Star Plate reader. Lysate protein concentrations were extrapolated from the BSA standard curve. 

### 2.14. Multiplex ELISA (MAGPIX) Assay

Retina protein lysates were diluted to 1 µg/µL in a NP-10 lysis buffer. The concentration of 32 cytokines and chemokines was measured using the Milliplex MAP Mouse Cytokine/Chemokine Magnetic Bead Panel (Millipore cat no. MCYTMAG70PMX32BK) as per the manufacturer’s instructions. Samples were read in duplicates using 25 µg total protein per well. Magnetic beads were incubated with the samples, standards, or quality controls overnight at 4 °C with constant agitation. Beads were washed as indicated by the manufacturer using a handheld magnetic plate. After incubation with the detection antibodies and Streptavidin-PE, the plate was analyzed using a Luminex MagPix System and the xPONENT software (v4.3, Luminex, Austin, TX, USA). Raw data was converted to concentrations using Milliplex Analyst software (v5.1, Millipore, St Louis, MO, USA). 

### 2.15. Western Blot

A total of 16 µg of total protein was mixed with 5× Laemmli buffer [34] containing 1 mM DTT and boiled for 5 min. Samples and 5 µL of Chameleon duo pre-stained protein ladder (Li-Cor) were loaded onto a Bolt 4–12% Bis-Tris Plus gel and ran at 200 V for 30 min using 1× Bolt MES SDS running buffer. The gel was then transferred into a nitrocellulose membrane using the iBlot 2 transfer system (Invitrogen) P0 program (23 V for 7 min). The membrane was then blocked in Intercept (TBS) blocking buffer (Li-Cor) for 45 min at room temperature. The membrane was then incubated with an anti-GFP antibody (Invitrogen cat no 33-2600, 1:500 dilution), and anti-beta Tubulin (Novus Biologicals cat no NB600-936, 1:1000 dilution), and 0.1% Tween-20 overnight at 4 °C with constant agitation. The membrane was then washed three times (5 min per wash) with PBS-T and then incubated with a goat anti-mouse IRDye 680RD (Li-Cor cat no 925-68070) and a goat anti-rabbit IRDye 800CW (Li-Cor cat no 925-32211) both diluted 1:5000 in Intercept (TBS) blocking buffer containing 0.1% Tween-20 for 45 min at room temperature. The membrane was washed as done before and then scanned using a CLx Odyssey Scanner (Li-Cor). The color image was converted into an 8-bit black and white image afterwards using ImageJ software (v1.5r, NIH, Bethesda, MD, USA). 

### 2.16. IL-21 Enzyme Linked Immunosorbent Assay (ELISA)

A mouse IL-21 ELISA kit from Peprotech (cat no 900-M368) was used according to the manufacturer’s protocol to determine the concentration of IL-21 in retina lysates. Briefly, all retina lysates were diluted in diluent (0.05% Tween 20, 0.1% BSA, 1× PBS) to 0.04 µg/µL. A capture antibody was reconstituted in deionized water to the recommended concentration and further diluted in PBS. Capture antibody was then aliquoted at 100 µL per well into a Nunc MaxiSorp flat bottom 96-well plate which was then covered with a film of parafilm and incubated over night at room temperature on a plate shaker. The next day the plate was washed three times with wash buffer (0.05% Tween 20 in PBS) followed by blotting on a paper towel. A total of 300 µL of blocking buffer (1% BSA in PBS) was added to each well. The plate was incubated at room temperature for 1 h on a plate shaker. After blocking, the plate was washed three times with wash buffer as described earlier. Each well then received 100 µL of either diluent, standard (0–3000 pg/mL of recombinant mouse IL-21), or lysate accordingly. The plate was sealed with a film and incubated at 4 °C overnight on a plate shaker. 

Samples and standards were removed after incubation and the plate was washed three times with wash buffer as described earlier. A total of 100 µL of diluted detection antibody (0.5 µg/mL) was added to each well and the plate was incubated at room temperature for 2 h on a plate shaker. The plate was washed again as done earlier, and 100 µL of Streptavidin-HRP (1:2000) was added to each well and the plate was incubated at room temperature again for 30 min on a plate shaker. The plate was washed as done previously, and 100 µL of ABTS substrate was added to each well. The plate was incubated at room temperature for 1 h on a plate shaker. Finally, the absorbance at 405 nm was measured using a Clario Star plate reader.

### 2.17. Statistical Analysis

Data comparing only three groups was analyzed by a one-way ANOVA followed by a post-hoc Tukey’s test to identify differences between groups. When only two sets of values were compared an independent student t-test was used to determine significance. Statistical significance was considered when *p*-value ≤ 0.05. Data were analyzed using the GraphPad Prism 8.2.1 statistical software (San Diego, CA, USA). A post hoc power calculation was performed using G*Power 3.1 software (Dusseldorf, Nordrhein-Westfalen, Germany) [35]. Significance labels in figures are defined as: * = *p*-value ≤0.05, ** = *p*-value ≤ 0.01, *** = *p*-value ≤ 0.001, **** = *p*-value ≤ 0.0001)

## 3. Results

### 3.1. Retinal Gene Delivery of a Secretable TatM013 Does Not Alter the Mouse Retina

#### 3.1.1. Effect of TatM013 Expression on Retina Function

To determine if the expression of TatM013 would affect the function of the retina, we intravitreally injected C57BL6J mice with 3 × 10^10^ vector genome copies of AAV2quad(Y-F)+T491V delivering either secretable GFP (sGFP) or sGFP-TatM013. These mice were then evaluated one month later using electroretinography (ERG). This technique stimulates the retina with a flash of light and measures the electrical response (change in potential) of the eye as a response to this stimulus. As demonstrated in Figure 1A, when the average amplitude of the a-wave (photoreceptors response), b-wave (interneurons response), or c-wave (RPE response) were recorded, there was no statistically significant difference between eyes treated with sGFP or sGFP-TatM013. We have previously compared the effect of GFP expression in the retina with that of a sham injection and demonstrated the lack of statistical significance between a sham injection and a GFP control injection [36]. These results suggest that the expression of TatM013 in the retina does not significantly affect the electrophysiological properties of the tissue.

#### 3.1.2. TatM013 Does Not Alter the Retina Structure

Another characteristic of the retina is its laminated structure which must be maintained for it to function. We tested the effects of TatM013 on the retina structure by spectral domain optical coherence tomography (SD-OCT), which utilizes infrared light to generate an image that represents the retina layers in a living animal. A total of 250 B-scans images were obtained and averaged into 25 high resolution vertical B-scans. Finally, we measured the thickness of each retina layer using an auto-segmentation software developed by Bioptigen. Our results showed no statistically significant difference between eyes expressing sGFP and eyes expressing TatM013 (Figure 1B). This observation suggests that retinal expression of TatM013 does not alter the structure of any of the retina layers.

### 3.2. Retinal Gene Expression of TatM013 Protects the Retina in an Experimental Autoimmune Uveitis (EAU) Mouse Model

#### 3.2.1. Retinal Expression of TatM013 Reduces the Clinical Signs of the EAU Mouse Model

We have previously demonstrated that retinal gene expression of TatM013 reduced the concentration of IL-1β and the recruitment of infiltrative cells to the retina and vitreous in a mouse model of endotoxin-induced uveitis [36]. One limitation of this model is that it does not involve the activation of autoreactive T-cells, which are known to be important for the development of uveitis in patients. Therefore, we now tested our TatM013 vector in a mouse model of experimental autoimmune uveitis in which autoreactive T-cells against the interphotoreceptor retinoid-binding protein (IRBP) are induced by immunizing B10.RIII mice with the IRBP_161–180_ peptide in complete Freund’s adjuvant (CFA). Mice were treated with an intravitreal injection of AAV2quad(Y-F)+T491V vector delivering either sGFP or sGFP-TatM013. One month later, gene expression was confirmed using fluorescence funduscopy detecting the expression of sGFP or sGFP-TatM013v5 in the retina (Figure 2A). Afterwards mice were immunized with the IRBP_161–180_ peptide to induce autoimmune uveitis. The peak of inflammation of this model occurs at 14 days post injection, at which time point we used fundoscopy to quantify the magnitude of retinal inflammation among the mice. After the evaluation, the mice retinas were harvested for protein or RNA extraction. Using western blot, we demonstrated that mice were still expressing sGFP or sGFP-TatM013v5 (Figure 2B).

Fundus images showed that mice treated with the sGFP-TatM013 had fewer signs of retinal inflammation when compared to sGFP treated eyes (Figure 3A–C). Fundus images from sGFP-treated eyes showed signs of vitreous hemorrhage and papilledema (Figure 3B red arrow heads). Furthermore, the eyes treated with the sGFP-TatM013 vector had fundus images resembling the eyes of mice injected intravitreally with saline but not immunized with the IRBP_161–180_. We then quantified the magnitude of retinal inflammation in the fundus images using a clinical score scale. Our results demonstrate that in fact, eyes treated with the sGFP-TatM013 vector had a clinical score indistinguishable from that of non-immunized animals (Figure 3D). These results demonstrate that the retinal expression of TatM013 abrogated the retinal inflammation associated with the EAU mouse model.

#### 3.2.2. TatM013 Decreases the Recruitment of Infiltrative Cells in the EAU Mouse Model

To determine if the retinal expression of TatM013 decreased the recruitment of infiltrative cells into the vitreous humor, we use SD-OCT to assess the structure of the retina in our mice. Using ImageJ software, we processed three images from each SD-OCT file and quantified the number of hyper reflective cells within the vitreous humor. The average values represent an estimate of the number of inflammatory cells that have invaded the retina. Figure 4A–C are representative SD-OCT images of eyes injected only with saline (A) and AAV-sGFP (B) or AAV-sGFP-TatM013 (C) followed by immunization with IRBP_161–180_ peptide. When values are averaged for each treatment, it was observed that eyes treated with the TatM013 vector exhibit a significantly decrease number of infiltrative cells when compared to sGFP treated eyes (Figure 4D). These results suggest that the retinal expression of TatM013 protects the retina from EAU inflammation by decreasing the number of retinal infiltrative cells.

To validate the hyper-reflective signals in the vitreous humor with histological observations, we embedded a small number of eyes in paraffin to generate sections stained with hematoxylin and eosin. Microscopic evaluation of these slides revealed the presence of inflammatory cells within the vitreous humor of sGFP treated eyes (Figure 5A–D). Similar inspection of the sGFP-TatM013 treated eyes showed significantly fewer infiltrative cells in the vitreous humor (Figure 5E–H). Furthermore, sGFP treated eyes had significant protein deposition in the anterior chamber (Figure 5A) and retina structural changes (Figure 5D), which were not observed in the sGFP-TatM013 treated eyes (Figure 5E,H). Overall, these indicate that TatM013 protects the retina by decreasing the number of infiltrative cells and decreasing the effects noted in ocular pathology.

### 3.3. M013 Modulates Inflammatory and Anti-Inflammatory Genes in the Retina of the EAU Mouse Model

#### 3.3.1. M013 Changes Gene Expression Pattern in the Retina of EAU Mice

To determine the anti-inflammatory effect of TatM013 in the retina, we isolated RNA from eyes treated with either sGFP or sGFP-TatM013 and used RT-qPCR to measure the gene expression differences between these groups. Retinas treated with the sGFP-TatM013 had significant increases in IL-27Ra and Ppar-γ, both of which are associated with anti-inflammatory macrophages [37,38], when compared to the sGFP treated retinas (Table 3). Furthermore, when we measured the expression of pro-inflammatory genes, we found that sGFP-TatM013 treated retinas had significantly lower expression of IL-17A and C3 when compared to sGFP treated retinas (Table 3). IL-17a is released by Th-17 cells, while localized C3 is produced by invading macrophage and activated microglia in response to retinal injury [39]. Together these results indicate that TatM013 expression in the retina can inhibit the expression of some pro-inflammatory pathways while increasing some anti-inflammatory signaling pathways in the setting of autoimmune uveitis. This overall response due to M013 expression highlights the complexity of NF-κB inhibition and warrants further investigation when considering the therapeutic value of this potent anti-inflammatory gene.

#### 3.3.2. M013 Modulates Multiple Cytokines and Chemokines in the Retina of the EAU Mouse Model

M013 is known to bind to the transcription factor NF-κB [6,8]. This interaction results in the sequestration of NF-κB within the cytoplasmic compartment which abrogates its function as a transcription factor. As a result, M013 expression significantly decreases the expression of NF-κB regulated genes such as IL-6. These observations come from in vitro studies in which THP-1 cells are transfected with a plasmid delivering M013 followed by their infection with a vMyx-M013KO virus [6]. We have previously demonstrated that our AAV-sGFP-TatM013 vector decreased the concentration of IL-1β in the vitreous humor of mice acutely challenged with intravitreal LPS [9]. Since IL-1β is regulated by both the NF-κB and the ASC-dependent activation of capsase-1, we decided to determine if other genes whose expression depend on NF-κB activity are inhibited in our EAU mouse model. Using a multiplex ELISA (MagPix) assay we quantify the concentration of 32 different cytokines and chemokines. The concentration of eight cytokines/chemokines was differentially regulated in sGFP-TatM013v5 when compared to sGFP (Figure 6A). Of note, the concentration of granulocyte-colony stimulating factor (G-CSF), Eotaxin/CCL11, and IL-6 (which are cytokines regulated by NF-κB [40,41,42]) were decreased in the sGFP-TatM013v5 treated eyes, thus suggesting the inhibition of the NF-κB transcription factor.

In addition to IL-17A and IFN-γ, another important cytokine in the EAU mouse model is IL-21 which is pleiotropic cytokine that belongs to the IL-2 family [43]. It is known that deletion of the IL-21 receptor abrogates the retinal damage of the EAU mouse of autoimmune uveitis [44]. To test if M013 expression in the retina affected the concentration of this cytokine, we performed an IL-21 ELISA on retinal lysates from our mice. Lysates from M013 expressing eyes had a 60% reduction in the concentration of IL-21 when compared to sGFP treated eyes (Figure 6B). This result suggests that the protective effects of M013 in the retina of the EAU mouse model was mediated in part by decreasing the levels of IL-21 in the retina.

## 4. Discussion

We have evaluated the use of an AAV vector that delivers a secretable and cell-penetrating form of the myxoma M013 gene. We have previously demonstrated that such vector can decrease ocular inflammation due to an intravitreal injection of lipopolysaccharide by decreasing IL-1β within the retina [9]. However, that mouse model of ocular inflammation (EIU) depends on the activation and recruitment of neutrophils and monocytes, which are not the prominent cells in the human disease. Thus, we used the experimental autoimmune uveitis mouse model to validate the anti-inflammatory properties of the M013 vector. We first demonstrated that retinal expression of M013 does not affect the retinal function (by ERG) or structure (by SD-OCT) in non-diseased animals. We then induced autoimmune uveitis by immunizing the B10.RIII mice with an IRBP_161–180_ peptide in CFA. Our evaluation of the mice retina demonstrated that M013 treated eyes had significantly diminished ocular inflammation when compared with sGFP treated eyes, which was corroborated by using a clinical score. Furthermore, when we quantified the number of infiltrating cells within the vitreous humor using SD-OCT images we found a significant decrease in the number of these cells. Gene expression studies in these animals showed that the expression of TatM013 caused an increase in the expression of anti-inflammatory genes simultaneously with a decrease in pro-inflammatory genes, especially IL-17A which is known to be a fundamental cytokine in the pathology of the EAU mouse model.

Our results demonstrate that the NF-κB and the inflammasome signaling within the retina are important in the development of uveoretinitis. Our gene expression studies showed a significant decrease in C3 and IL-17A which are potent pro-inflammatory molecules that are regulated by NF-κB activation [45,46]. Humanized anti-IL-17A antibody has been evaluated in the setting of non-infectious autoimmune uveitis showing clinical benefits when delivered intravenously at 10–30 mg/kg biweekly for two months [47]. Although important, this systemic approach could affect the ability of the patients to fight infections outside the eye due to its systemic delivery. In contrast our approach would minimize the systemic immunosuppressive effects by acting only in the affected tissue without the need of a frequent intervention. Furthermore, since the expression of the therapeutic M013 will be constant we expect that our gene therapy approach would significantly decrease the number of inflammatory flares in patients with recurrent uveitis when compared to the current standard of care.

The broad effect of M013 expression in the retina of EAU mice observed by qRT-PCR and multiplex ELISA suggests that M013 can indirectly modulate multiple immune relevant pathways in the retina. This could possibly be the ability of M013 to modulate the expression of cytokines involved in the early stages of the disease or due to a yet uncharacterized interaction of M013 with proteins expressed within retinal cells. A better understanding of this intricate effect will allow the refinement of our efforts to translate the use of M013 as a therapeutic intervention. The recent characterization of M013 interaction with either NF-κB or ASC-1 by Garg et al. [8] opens the possibility to determine the contribution of each pathway towards the overall protective effect of this gene in the setting of EAU. We are currently testing some of these mutant forms of M013 in other animal models. Finally, the use of single cell RNAseq will allow the characterization of the cellular and gene expression changes mediated by M013 expression in the retina of the EAU mouse model, thus allowing the identification of potential side effects due to prolonged inhibition of these pathways in the retina.

A limitation of our study is that it only demonstrated that AAV-TatM013 protects against retinal inflammation, but it did not evaluate if this vector can also treat the disease. Due to the time it takes for the inflammation to peak in the EAU mouse model and the time it takes for the AAV to express a transgene in the post-mitotic cells of the retina (~21–30 days), it is not feasible to test if this vector can treat uveoretinitis using this model. Future experiments will combine the use of self-complementary AAV vectors which are known to have a shorter time to expression in the retina (~14 days) with the use of a mouse model of spontaneous autoimmune uveitis. The B10.RIII mice carrying the R161M T-cell receptor mutation was developed and characterized by Horai et al. and developed spontaneous autoimmune uveitis with a 100% incidence by 11 weeks of age [16,29]. This mouse model will allow the testing once the disease commences and to determine how M013 affects the progression of the disease.

## 5. Conclusions

We have demonstrated that the use of an AAV delivering a secretable and cell penetrating M013 gene can protect the eye against ocular inflammation due to autoimmune damage. This viral vector can be of benefit in autoimmune diseases such as rheumatoid arthritis or sympathetic ophthalmia.

## 6. Patents

This work is covered in US patent application US20160376325A1.

## Figures and Tables

**Figure 1 jcm-08-02082-f001:**
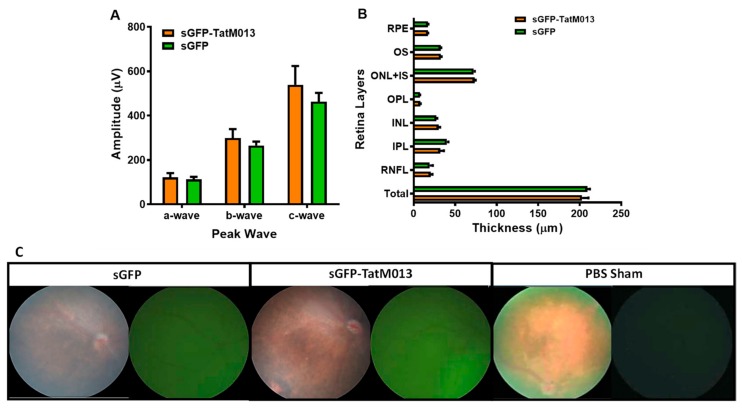
Retinal expression of TatM013 does not alter the retina function or structure. (**A**) Eight-week old mice were injected intravitreally with 3 × 10^10^ vgc of AAV2quad(Y-F)+T491V vector delivering either sGFP-TatM013 or sGFP. Mice were evaluated by electroretinogram one-month post-injection. Average amplitudes of a-wave, b-wave, and c-wave were recorded. No statistically significant difference in amplitude was observed between eyes treated with either sGFP-TatM013 or sGFP vector. (**B**) The thickness of the different retinal layers was measured using spectral domain optical coherence tomography. Auto-segmentation software was utilized to obtain measurements of the different retina layers. No statistically significant difference was observed between eyes expressing sGFP-TatM013 or sGFP. (**C**) Retina fundus images injected with sGFP, sGFP-TatM013, or phosphate buffered saline (PBS) sham. Fluorescence images show a diffused pattern of GFP which is absent in PBS sham injected mice. Values are reported as average +/- SEM. (*n* = 5 mice).

**Figure 2 jcm-08-02082-f002:**
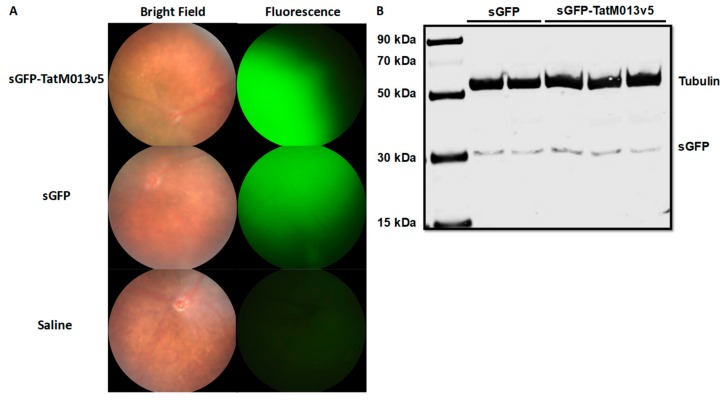
Expression of AAV transgene before and after experimental autoimmune uveoretinitis (EAU) induction. (**A**) B10.RIII mice were evaluated by funduscopy one month after an intravitreal injection of AAV delivering sGFP-TatM013v5 or secreted GFP (sGFP). As a control we evaluated mice that received an intravitreal injection of saline. Diffuse fluorescence indicates the secretion of sGFP and sGFP-TatM013v5 in the retina. (**B**) Western blot from retinas harvested 14 days after IRBP immunization. Membrane was probed with anti-GFP and anti-Tubulin antibodies. Image shows expression of sGFP on both sGFP and sGFP-TatM013v5 retina lysates. (*n* = 2–3 retina samples from different mice per group).

**Figure 3 jcm-08-02082-f003:**
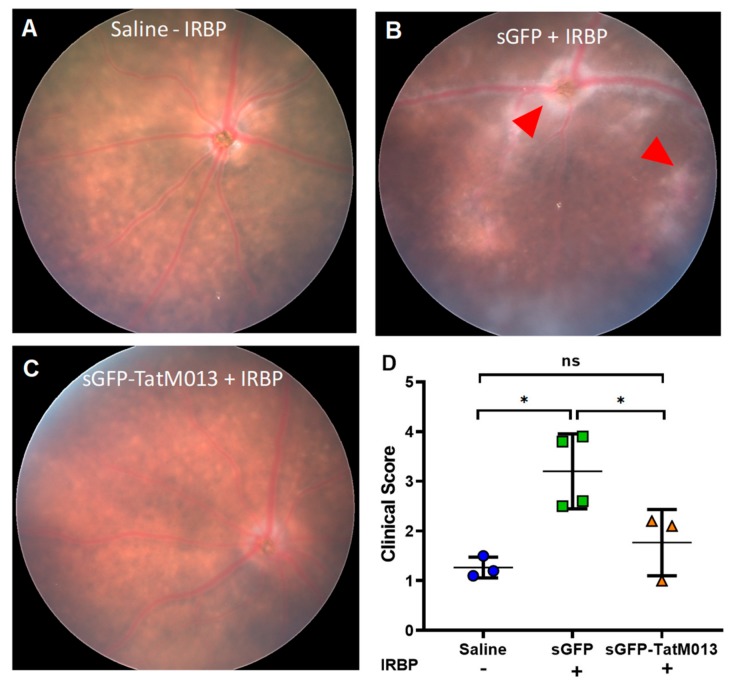
TatM013 decreases the clinical score of the experimental autoimmune uveoretinitis (EAU) mouse model. Fundus images of mice injected intravitreally with (**A**) saline, (**B**) AAV-sGFP, or (**C**) AAV-sGFP-TatM013 fourteen days post immunization with PBS (-IRBP) (**A**) or IRBP peptide (+IRBP) (**B**,**C**). (**D**) Fundus clinical score quantification. Values reported as average +/- standard deviation. (*n* = 4-3 mice, power (1-β) =0.9456, * =*p*-value ≤ 0.05). AAV, adeno associated virus; sGFP, secreted GFP; PBS, phosphate buffered saline; IRBP, interphotorecptor retinoid binding protein.

**Figure 4 jcm-08-02082-f004:**
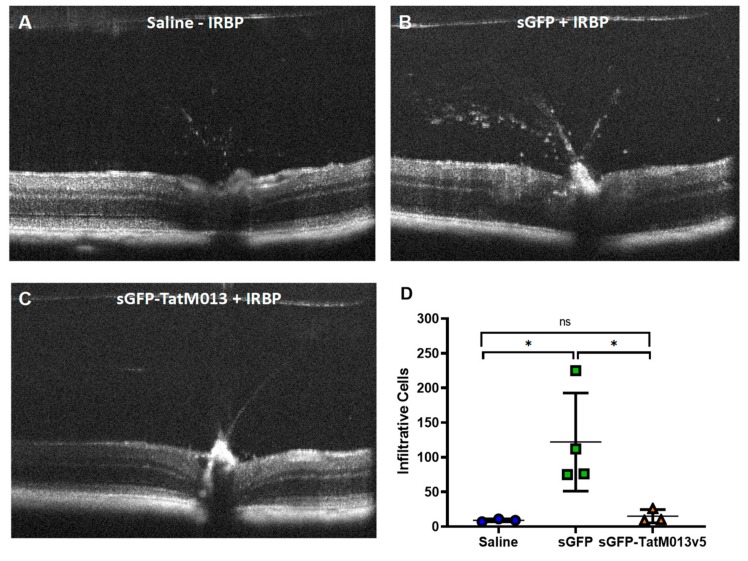
Retinal expression of TatM013 decreases the number of infiltrative cells in the experimental autoimmune uveoretinitis (EAU) mouse model. Images of the retina and vitreous humor were taken using SD-OCT. Representative B-scans of mice injected with (**A**) saline, (**B**) AAV-sGFP, and (**C**) AAV-sGFP-TatM013 fourteen days after treatment with (**A**) PBS or (**B**,**C**) IRBP peptide. (**D**) The numbers of infiltrative cells within the vitreous humor were quantified using ImageJ. Values are reported as average +/- standard deviation. (*n* = 4-3 mice, power (1-β) = 0.9944, *= *p*-value ≤ 0.05). AAV, adeno associated virus; sGFP, secreted GFP; SD-OCT, spectral domain optical coherence tomography; PBS, phosphate buffered saline; IRBP, interphotorecptor retinoid binding protein.

**Figure 5 jcm-08-02082-f005:**
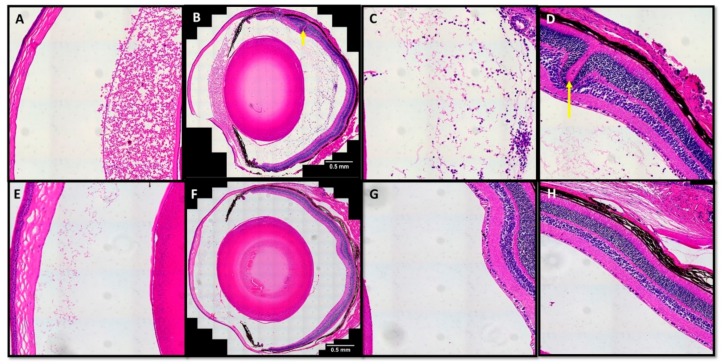
TatM013 improves the histopathological hallmarks of experimental autoimmune uveoretinitis (EAU). (**A–D**) Eyes from secreted GFP (sGFP) treated mice, or (**E–H**) sGFP-TatM013 treated mice were enucleated, fixed in 4% paraformaldehyde and embedded in paraffin. Four sections from each eye were captured on a slide and stained with hematoxylin and eosin. The anterior chamber of the sGFP treated eyes had a significant amount of proteinaceous material deposited (**A**) that was absent from sGFP-TatM013 treated eyes (**E**). There was also an increase in number of inflammatory cells in the vitreous body of the sGFP treated eyes (**B****,C**) when compared to sGFP-TatM013 treated eyes (**F–G**). Retina rosettes (yellow arrow) and thinning can be seen in sGFP (**D**) but not in sGFP-TatM013 treated eyes.

**Figure 6 jcm-08-02082-f006:**
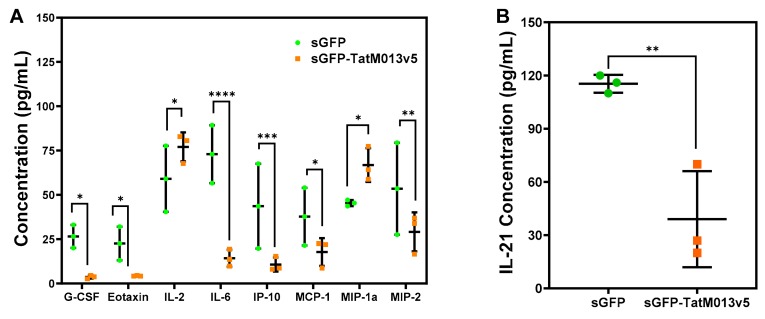
TatM013v5 expression decreases the concentration of multiple cytokines/chemokines in the retina of the experimental autoimmune uveoretinitis (EAU) mouse model. (**A**) Protein lysates from secreted GFP (sGFP) or sGFP-TatM013v5 treated retinas were diluted in NP-10 lysis buffer to 1 µg/µL of total protein concentration. A multiplex ELISA assay was used to measure the concentration of 32 cytokines and chemokines. Nine analytes were differentially regulated in the sGFP-TatM013v5 lysates. Retina lysates from sGFP-TatM013v5 had a statistically significantly lower concentration of granulocyte-colony stimulating factor (G-CSF), Eotaxin (CCL11), interleukin 6 (IL-6), interferon gamma-induced protein 10 (IP-10, CXCL10), monocyte chemoattractant protein 1 (MCP-1), and macrophage inflammatory protein 2 (MIP-2) when compared to sGFP retina lysates. (**B**) An IL-21 ELISA was performed to quantify the concentration of IL-21 in retina lysates. Retina lysates from sGFP-TatM013v5 eyes had statistically significant lower concentration of IL-21 when compared to sGFP lysates. Values are reported as average +/- standard deviation. (*n* = 3 retina lysates from three different mice). * = *p*-value ≤0.05, ** = *p*-value ≤ 0.01, *** = *p*-value ≤ 0.001, **** = *p*-value ≤ 0.0001.

**Table 1 jcm-08-02082-t001:** Oligonucleotides used for RT-qPCR.

Gene	Forward Oligo (5′-3′)	Reverse Oligo (5′-3′)
*Arg1*	AAGACAGGGCTCCTTTCAGG	CTGTGATGCCCCAGATGGTT
*C3*	CAGAAACGCCCTGAAGCTG	CAGTTGGGACAACCATAAAC
*Fcgr1*	CAAGTGCTTGGTCCCCAGTC	ACACGCCATCGCTTCTAACT
*ICAM-1*	AAGGTGGTTCTTCTGAGCGG	TCCAGCCGAGGACCATACAG
*IL-17A*	GGAGAGCTTCATCTGTGTCTCTG	ACTTTTGCGCCAAGGGAGTT
*IL27ra*	TGCAGGAACGTGAGCAGTC	CCCATCAGAAGCAAACCCCA
*Marco*	ATCCAGGGATTGCAGGTGTG	CTGGCCAGAAGACCCTTTCA
*Mrc1*	TTGCACTTTGAGGGAAGCGA	AGTCCAATCCAGAGTCCCGA
*Pparg*	CGCTGGGGTATTGGGTCG	TTTCAAATCTTGTCTGTCACACAGT
*Retn1a*	CCCCAGGATGCCAACTTTGA	CAGTAGCAGTCATCCCAGCA
*Slamf1*	GAGCTTCTTCCTTGGGGGTAA	CCATCACACCTCCACCTGTTC
*SOCS3*	CCCTTCCCGGCCCAG	CGGGGAGCTAGTCCCGAA
*STAT1*	TGCAGTGAGTGAGTGAGAGC	GAACCACTGTGACATCCTTGAG
*STAT6*	GCTACTGGTCAGATCGGCTG	CAGTGAGCGAATGGACAGGT
*TGF-β*	GCCTGAGTGGCTGTCTTTTG	TTTGGGGCTGATCCCGTTG
*β-Actin*	CTGTCGAGTCGCGTCCACC	ATTCCCACCATCACACCCTGG

**Table 2 jcm-08-02082-t002:** RT-qPCR conditions.

Temperature (°C)	Time (min)	Cycles
95.0	0:30	1
95.0	0:15	40
60.0	0:30	
65.0	0:05	1
0.5 increases		
95.0		

**Table 3 jcm-08-02082-t003:** Retina gene expression changes in sGFP-TatM013 and sGFP treated eyes.

Gene	sGFP			sGFP-TatM013v5			Multiple T-Test		
	Average	± StDev	*n*	Average	± StDev	*n*	Difference	*p*-Value	Significance
*Arg1*	1.00	0.11	3	1.75	0.77	3	0.75	0.17022	ns
*C3*	1.00	0.10	3	0.20	0.10	3	−0.8	0.000608	***
*Fcgr1*	1.00	0.83	3	0.40	0.34	3	−0.6	0.311063	ns
*ICAM-1*	1.00	0.30	3	1.50	0.30	3	0.5	0.110787	ns
*IL-17A*	1.00	0.30	3	0.20	0.10	3	−0.8	0.011858	*
*IL-27Ra*	1.00	0.58	3	5.19	0.73	3	4.19	0.001469	***
*Marco*	1.00	0.14	3	0.91	0.64	3	−0.09	0.823617	ns
*Mrc1*	1.00	0.15	3	0.65	0.52	3	−0.35	0.325368	ns
*Ppar-*γ	1.00	0.97	3	15.29	0.78	3	14.29	0.000038	****
*Retn1a*	1.00	0.09	3	1.21	0.53	3	0.21	0.535767	ns
*Slamf1*	1.00	0.86	3	0.37	0.57	3	−0.63	0.349871	ns
*Socs3*	1.00	0.09	3	0.56	0.33	3	−0.44	0.089808	ns
*Stat1*	1.00	0.84	3	1.01	0.61	3	0.01	0.987487	ns
*Stat6*	1.00	0.42	3	1.23	0.84	3	0.23	0.69325	ns
*TGF-β*	1.00	0.50	3	0.40	0.20	3	-0.6	0.125844	ns

StDev = standard deviation, ns = not significant. * = *p*-value ≤0.05, *** = *p*-value ≤ 0.001, **** = *p*-value ≤ 0.0001.

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
