# Peer review of "AAV Mediated Delivery of Myxoma Virus M013 Gene Protects the Retina against Autoimmune Uveitis"

_jcm, 2019, doi:10.3390/jcm8122082_

Round 1
Reviewer 1 Report
Ridley et al. present a novel gene delivery approach to protecting the retina against autoimmune uveitis. An AAV vector was used to deliver a secreted, cell-targeted myxoma virus NFkB/ASC-1 inhibitor to the retina prior to vaccination which induces auto-reactive T cells and retinal inflammation.
While a clear reduction in clinical score, cellular infiltrate and histopathology is observed in the retina of mice delivered the viral fusion, it is important to demonstrate that the AAV vector itself is safe, and that the inhibitor is being produced at the time of vaccination and retinal inflammation, and its effects are mediated through the inhibition of inflammasome and NF-kB signalling.
Critical issues to be addressed:
The study claims that the vector is safe, when the retinal structure and function analyses only show that addition of M013 has no effect relative to GFP delivery alone. Both AAV vectors should be compared to mock-infected retina, so as to prove the delivery method itself is safe. The study also claims that the clinical effects observed are due to the actions of the secretable TatM013. To make this claim, the retinal expression of the fusion protein at the time of vaccination, or at the peak of the autoimmune response must be confirmed. This could readily be achieved using existing samples through staining of paraffin-embedded sections for GFP, or by qRT-PCR analysis using GFP or M013-specific primers. Another claim is that the clinical effects are due to the NFkB/ASC-1 inhibitory actions of the secretable TatM013. While qRT-PCR analysis show changes in immune markers, they do not provide evidence of reduced inflammation. Again this could readily be achieved using existing samples through staining of paraffin-embedded sections for cleaved IL-1b, or by qRT-PCR or ELISA analysis of NFkB-specific gene products. Even if published protocols are used, the methods for production of the AAV vectors, and their infection of the retina, and for IRBP vaccination need to be briefly described. The statistical analyses need to be revised. For analyses involving three treatment groups at a single timepoint (Fig 2-3), one-way ANOVA should be used. For qRT-PCR analysis of two treatment groups at a single time point (Table 3), t-tests should be used, not ANOVA as indicated.
Additional recommendations:
It would be helpful for the reader to provide a rational for trialing a secreted cell-targeted form of M013, as opposed to just delivering native M013. It would also help the reader to introduce human autoimmune uveitis and its pathogenesis early on, referring to critical cells (ie. macrophages and Th17 cells) and regulatory molecules (IL-17, IL-27, IL-21, TGFb etc) involved in its induction. This will help justify why they are studied here. The experimental model used in the study could also be introduced earlier, rather than being justified in the results. Quantification of histo-pathological hallmarks presented in Figure 4, and graphs showing the mean for each treatment group, would be a valuable addition. Immunostaining for macrophage and T cell subtypes within the paraffin-embedded sections would validate and help interpretation of the qRT-PCR data. Table 3 shows the Avg TGF-b reading for sGFP group to be 1.10 when it should be 1.0. Many autoimmune diseases suffer from flare-ups, providing a window for preventative treatments. Is this the case for autoimmune or other inflammatory diseases of the retina?
Author Response
Critical issues to be addressed:
The study claims that the vector is safe, when the retinal structure and function analyses only show that addition of M013 has no effect relative to GFP delivery alone. Both AAV vectors should be compared to mock-infected retina, so as to prove the delivery method itself is safe.
ANSWER: We thank the reviewer for this suggestion. We have previously published data using a different AAV vector which demonstrates that compared to a sham injected animal, the intravitreal injection of a GFP control AAV does not cause a significant difference in ERG or OCT readings (Ildefonso et al 2015). We have modified our manuscript to indicate the absence of a difference between our AAV injected animals and the reported sham injected mice.
The study also claims that the clinical effects observed are due to the actions of the secretable TatM013. To make this claim, the retinal expression of the fusion protein at the time of vaccination, or at the peak of the autoimmune response must be confirmed. This could readily be achieved using existing samples through staining of paraffin-embedded sections for GFP, or by qRT-PCR analysis using GFP or M013-specific primers.
ANSWER: We thank the reviewer for this suggestion. In response we now include fundus images from our animals taken the day of immunization and at the peak day of inflammation showing green fluorescence which indicates the presence of secreted GFP. We have also included western blot results demonstrating the presence of both secreted GFP and TatM013v5 in retina lysates obtained at the day of peak inflammation. We know these results demonstrate that our transgene is being expressed in the retinas of these animals.
Another claim is that the clinical effects are due to the NFkB/ASC-1 inhibitory actions of the secretable TatM013. While qRT-PCR analysis show changes in immune markers, they do not provide evidence of reduced inflammation. Again this could readily be achieved using existing samples through staining of paraffin-embedded sections for cleaved IL-1b, or by qRT-PCR or ELISA analysis of NFkB-specific gene products.
ANSWER: We now include concentration readings for G-CSF, Eotaxin (CCL11), and IL-6 (figure 6) which are cytokines known to be regulated by the NF-kB transcription factor.
Even if published protocols are used, the methods for production of the AAV vectors, and their infection of the retina, and for IRBP vaccination need to be briefly described.
ANSWER: We have added some of the requested methods.
The statistical analyses need to be revised. For analyses involving three treatment groups at a single timepoint (Fig 2-3), one-way ANOVA should be used. For qRT-PCR analysis of two treatment groups at a single time point (Table 3), t-tests should be used, not ANOVA as indicated.
ANSWER: We thank the reviewer for the suggestion. We have modified our analysis of the qRT-PCR analysis using a multiple t-test approach and report the new p-values. This approach did not identify TGF-β as a statistically significant differently expressed gene, thus we have modified our conclusions as to correlated with our new statistical analysis.
Additional recommendations:
It would be helpful for the reader to provide a rational for trialing a secreted cell-targeted form of M013, as opposed to just delivering native M013. It would also help the reader to introduce human autoimmune uveitis and its pathogenesis early on, referring to critical cells (ie. macrophages and Th17 cells) and regulatory molecules (IL-17, IL-27, IL-21, TGFb etc) involved in its induction. This will help justify why they are studied here.
ANSWER: We thank the reviewer for this suggestion. We have modified our introduction (lines 62-72) to discuss human autoimmune uveitis and its pathogenesis.
The experimental model used in the study could also be introduced earlier, rather than being justified in the results.
ANSWER: We have now added a description of the EAU mouse models in the introduction (lines 84-94) as a response to this comment.
Quantification of histopathological hallmarks presented in Figure 4, and graphs showing the mean for each treatment group, would be a valuable addition.
ANSWER: We thank the reviewer for this suggestion. However, we do not have sufficient number of blocks to get reliable quantification and repeating the experiment would require at more time than the allowed to reply to this revision. We will include this endpoint for our future experiments.
Immunostaining for macrophage and T cell subtypes within the paraffin-embedded sections would validate and help interpretation of the qRT-PCR data.
ANSWER: We agree with the reviewer that this would be great data to add to our manuscript. However due to the time constraints, we will not be able to complete these experiments before we are required by the journal to submit a revised paper. We will include these studies in future experiments and use flowcytometry to quantify different cell types.
Table 3 shows the Avg TGF-b reading for sGFP group to be 1.10 when it should be 1.0.
ANSWER: We thank the reviewer for this observation. We have now corrected our oversight.
Many autoimmune diseases suffer from flare-ups, providing a window for preventative treatments. Is this the case for autoimmune or other inflammatory diseases of the retina?
ANSWER: As suspected by the reviewer, there are flares of retina inflammation in autoimmune uveitis. This provides the opportunity for exploring the benefits of gene therapy in the context of decreasing the number of flares a patient would have compared to standard of care treatment. We have modified our discussion (lines 428-430) to indicate this potential benefit of our AAV vector.
Reviewer 2 Report
The paper is well written and demonstrates decreased inflammation is associated with intravitreous M013 delivery by AAV in this model. I have only a few questions and suggestions.
Introduction
Please provide a little more information about the AAV construct used in this study. I realize that the 2014 Human Gene Therapy paper discusses this, but a more thorough review of that data would be beneficial for the introduction of this paper. Also, I suggest that you reword Line 69-70. You have not proven long-term anti-inflammatory effects with this construct. Maybe change to "This approach is expected to provide local long-term...". And you might want to explicitly state why you expect that outcome.
Materials & Methods
Materials and methods are very detailed (which is nice), but there are a few things missing and other things that may be unnecessary to include. The AAV construct development or acquisition of this construct should be mentioned in M&M. Also, a brief description of AAV treatment of the mice is in the abstract of the paper but is not in the methods section. Please provide a detailed description of how the mice were treated with AAV. Try to shorten other sections of M&M a bit. If you can say things were done per manufacturer's protocol and then just detail any deviations from the protocol, you should do that.
Results
Please describe the AAV vector you are using. The results section is the first time you mention its full designation.
Please be sure to include the numbers of mice used for each experiment.
Some of the information in the results section is repetitive (e.g. you describe that the scoring system for the retinal disease was blinded in M&M and at least a couple of times in the results). Try to minimize repetitive information.
If possible, point out rosettes and thinning of the retina in Figure 4 a little more clearly (arrows may help).
Please include a reference for Lines 281-282.
Author Response
Introduction
Please provide a little more information about the AAV construct used in this study. I realize that the 2014 Human Gene Therapy paper discusses this, but a more thorough review of that data would be beneficial for the introduction of this paper.
ANSWER: We have now added a description of the AAV-sGFP-TatM013 vector described in the 2014 Human Gene Therapy paper (lines 45-55, and 111-116).
Also, I suggest that you reword Line 69-70. You have not proven long-term anti-inflammatory effects with this construct. Maybe change to "This approach is expected to provide local long-term...". And you might want to explicitly state why you expect that outcome.
ANSWER: We thank the reviewer for the suggestion. We have now modified the sentence to read as follows: “This approach is expected to provide local long-term anti-inflammatory effects after a single injection based on the long-term effects described for AAV-RPE65 in LCA patients [27].”.
Materials & Methods
Materials and methods are very detailed (which is nice), but there are a few things missing and other things that may be unnecessary to include. The AAV construct development or acquisition of this construct should be mentioned in M&M.
ANSWER: We have added a section on the AAV construct development in the M&M section. However as requested by reviewer 1 we have decided to keep some of the detailed descriptions in the M&M section.
Also, a brief description of AAV treatment of the mice is in the abstract of the paper but is not in the methods section. Please provide a detailed description of how the mice were treated with AAV.
ANSWER: As requested we have now added a description of the AAV injections (lines 129-139) in the M&M section.
Try to shorten other sections of M&M a bit. If you can say things were done per manufacturer's protocol and then just detail any deviations from the protocol, you should do that.
ANSWER: We thank the reviewer for this suggestion. However as requested by reviewer 1 we have decided to keep most of the methods descriptions, with only minor changes.
Results
Please describe the AAV vector you are using. The results section is the first time you mention its full designation.
ANSWER: We have now added a description of the AAV vector serotype used in our studies in the M&M section (lines 117-119).
Please be sure to include the numbers of mice used for each experiment.
ANSWER: The number of animals is indicated in each figure.
Some of the information in the results section is repetitive (e.g. you describe that the scoring system for the retinal disease was blinded in M&M and at least a couple of times in the results). Try to minimize repetitive information.
ANSWER: We have modified the text to avoid repetition in this version of the manuscript.
If possible, point out rosettes and thinning of the retina in Figure 4 a little more clearly (arrows may help).
ANSWER: We have now modified our image by adding a yellow arrow to indicate the rosettes in the histological images (Figure 5).
Please include a reference for Lines 281-282.
ANSWER: We have now provided references for the mentioned lines (current lines 366-368).
Reviewer 3 Report
In this study, Ridley R, et al. tested a therapeutic efficacy of AAV-based gene delivery of myxoma virus M013 gene that is fused to secreted GFP (sGFP) and Tat peptide sequence, against autoinflammatory uveitis. This idea is based on their previous study reporting that the M013 gene expression protects retina in a nonspecific inflammation, endotoxin induced uveitis (EIU). In retina of healthy control B6 mice, M013 could not alter function and structure of the organ, compared with the secreted GFP (sGFP). In retina of experimental autoimmune uveoretinitis (EAU) mice, M013 expression significantly decreased disease score and the number of infiltrative cells in retina. M013 also suppressed induction of several host proinflammatory molecules that include IL-17A and IL-21. Based on these observations, the authors concluded that M013 gene delivery could be of clinical benefit against autoimmune diseases. The potential of this study is that M013 expression could suppress induction of autoinflammation, such as EAU that is mediated by autoreactive T cell reaction. The main concerns were…
In Fig.3 and 4, the data showed that M013 suppressed inflammatory cell infiltration. The infiltrative cells consist of several cell types such as neutrophil, monocytes as well as T cells. As mentioned by the authors, especially T cell is concerned in the human disease. Please clarify the percentage of each cell type in the infiltrative cells by immunostaining or suitable method.
In Fig.5, the author showed that M013 expression decreased IL21 in retina of EAU mouse model. Actually, this observation suggests a reduction on the number of cells secreting IL21 in retina of EAU mice. However, the detail mechanism of the reduction is unclear. There are two possibilities. The first is that M013 directly suppressed IL21 production of source cells (mainly T cells) in retina. The second is that M013 suppressed chemokine production of non-T cells, resulting in the reduction of the T cell infiltration from draining lymph node to retina. The information of M013 effect on expression of several chemokines, which associate with Th17 migration, especially CCL20 (PMID: 18025126), in retina should support our discussion about M013’s therapeutic effect against EAU.
In table 3, the authors demonstrated that M013 significantly upregulated IL-27Ra or Ppar-gamma gene expression. As mentioned in the manuscript, M013 function mainly depends on its inhibitory function against host transcription factors. Based on these observations, M013’s effect should be complex in vivo and this would be important if we consider M013 as therapeutic agent. Please discuss about this point.
L287, “TGF-beta, an anti-inflammatory cytokine…”. This description leads misunderstanding of readers about TGF-beta immune function. TGF-beta should associate with immunological tolerance as well as Th17 differentiation (PMID: 18469800).
In Fig.1, AAV-sGFP was explained as “GFP”. In the other figures, as sGFP. Please correct abbreviation.
Author Response
In Fig.3 and 4, the data showed that M013 suppressed inflammatory cell infiltration. The infiltrative cells consist of several cell types such as neutrophil, monocytes as well as T cells. As mentioned by the authors, especially T cell is concerned in the human disease. Please clarify the percentage of each cell type in the infiltrative cells by immunostaining or suitable method.
ANSWER: We thank the reviewer for this suggestion. Unfortunately, we do not have enough samples to obtain a statistically significant value. We will include your suggestion as an endpoint in our future experiments by quantifying cell types using flow cytometry. However, we would like to point out that studies conducted in a rat model of EAU suggest an increase number of T-cells (both CD4+ and CD8+) and B-cells are expected to be present in the vitreous humor of our mouse model (Pepple et al 2018, IOVS, 59(6):2504-2511).
In Fig.5, the author showed that M013 expression decreased IL21 in retina of EAU mouse model. Actually, this observation suggests a reduction on the number of cells secreting IL21 in retina of EAU mice. However, the detail mechanism of the reduction is unclear. There are two possibilities. The first is that M013 directly suppressed IL21 production of source cells (mainly T cells) in retina. The second is that M013 suppressed chemokine production of non-T cells, resulting in the reduction of the T cell infiltration from draining lymph node to retina. The information of M013 effect on expression of several chemokines, which associate with Th17 migration, especially CCL20 (PMID: 18025126), in retina should support our discussion about M013’s therapeutic effect against EAU.
ANSWER: We appreciate the reviewer’s suggestion. Although we did not measure CCL20, in this version of our manuscript we provide quantification of more cytokines and chemokines in figure 6. This was performed using a multiplex ELISA method that allowed us to screen 32 molecules, out of which we report those that had a statistically significant difference between the sGFP and sGFP-TatM013v5 groups. We believe that as suggested by the reviewer, M013 affects multiple chemokines that could lead to a decrease in IL-21.
In table 3, the authors demonstrated that M013 significantly upregulated IL-27Ra or Ppar-gamma gene expression. As mentioned in the manuscript, M013 function mainly depends on its inhibitory function against host transcription factors. Based on these observations, M013’s effect should be complex in vivo and this would be important if we consider M013 as therapeutic agent. Please discuss about this point.
ANSWER: We have now addressed this suggestion in our discussion (lines 430-441).
L287, “TGF-beta, an anti-inflammatory cytokine…”. This description leads misunderstanding of readers about TGF-beta immune function. TGF-beta should associate with immunological tolerance as well as Th17 differentiation (PMID: 18469800).
ANSWER: We have removed this statement from our manuscript given our new statistical analysis of the qRT-PCR data.
In Fig.1, AAV-sGFP was explained as “GFP”. In the other figures, as sGFP. Please correct abbreviation.
ANSWER: We have now corrected this discrepancy in abbreviations.
Round 2
Reviewer 1 Report
Ridley et al. present a novel gene delivery approach, which delivers a secreted cell-targeted myxoma virus inhibitor to the retina of mice, protecting the tissue from autoimmune uveitis induced via IRBP vaccination.
The authors provide strong evidence to support their conclusions that the AAV vector is safe, that the inhibitor is being produced in the retina, and that its presence is associated with a reduction in inflammasome and NF-kB signalling, and a reduction is clinical score, cellular infiltrate and histopathology.
Minor corrections needed:
In Figure 1, a label for part C is needed, and boxes obscuring the titles for sGFP and sGFP-TatM013 in part C should be removed.
Methods refer to both a Multiplex ELISA and to a IL-21 ELISA kit, both of which were performed on retinal lysates. The text on Page 13 refers to an IL-21 ELISA but links this with multiplex data in Figure 6. Please clarify which data set this refers to: either by providing IL-21 ELISA kit data as Figure 7, or by deleting the methods referring to the ELISA kit.
Author Response
In Figure 1, a label for part C is needed, and boxes obscuring the titles for sGFP and sGFP-TatM013 in part C should be removed.
ANSWER: We thank the reviewer for the observation. This is an alteration caused by the conversion of the word document into a pdf file. We have used an alternative method to convert our file and generated a pdf file without this labeling problem.
Methods refer to both a Multiplex ELISA and to a IL-21 ELISA kit, both of which were performed on retinal lysates. The text on Page 13 refers to an IL-21 ELISA but links this with multiplex data in Figure 6. Please clarify which data set this refers to: either by providing IL-21 ELISA kit data as Figure 7, or by deleting the methods referring to the ELISA kit.
ANSWER: We thank the reviewer for this observation. We did perform a multiplex ELISA using a kit which did not included IL-21. The remaining lysate was then used in a typical ELISA measuring IL-21. In response to the reviewer, we have plotted the results for IL-21 on Figure 6B instead of making them Figure 7. We believe these results should remain together as they pertain to the cytokines changes taking place in the retina of the EAU mice studied.
Reviewer 3 Report
The reviewer is overall satisfied with the changes made to the revised manuscript. In Fig.1C, Both left and middle panels lack figure title maybe due to electric file exchanges. Please check that.
Author Response
The reviewer is overall satisfied with the changes made to the revised manuscript. In Fig.1C, Both left and middle panels lack figure title maybe due to electric file exchanges. Please check that.
ANSWER: Thanks to the reviewer, we have now corrected this issue.